# Statistical Methods for Modeling the Compressive Strength of Geopolymer Mortar

**DOI:** 10.3390/ma15051868

**Published:** 2022-03-02

**Authors:** Hemn Unis Ahmed, Aso A. Abdalla, Ahmed S. Mohammed, Azad A. Mohammed, Amir Mosavi

**Affiliations:** 1Civil Engineering Department, College of Engineering, University of Sulaimani, Kurdistan Region, Sulaimaniyah 46001, Iraq; aso.abdalla@univsul.edu.iq (A.A.A.); ahmed.mohammed@univsul.edu.iq (A.S.M.); azad.mohammed@univsul.edu.iq (A.A.M.); 2Department of Civil Engineering, Komar University of Science and Technology, Kurdistan Region, Sulaimaniyah 46001, Iraq; 3Institute of Information Engineering, Automation and Mathematics, Slovak University of Technology in Bratislava, 81107 Bratislava, Slovakia; 4Institute of Information Society, University of Public Service, 1083 Budapest, Hungary; 5John von Neumann Faculty, Obuda University, 1034 Budapest, Hungary

**Keywords:** geopolymer, mortar, fly ash, geopolymer concrete, alkaline activator, prediction, compressive strength, machine learning, regression, construction materials

## Abstract

In recent years, geopolymer has been developed as an alternative to Portland cement (PC) because of the significant carbon dioxide emissions produced by the cement manufacturing industry. A wide range of source binder materials has been used to prepare geopolymers; however, fly ash (FA) is the most used binder material for creating geopolymer concrete due to its low cost, wide availability, and increased potential for geopolymer preparation. In this paper, 247 experimental datasets were obtained from the literature to develop multiscale models to predict fly-ash-based geopolymer mortar compressive strength (CS). In the modeling process, thirteen different input model parameters were considered to estimate the CS of fly-ash-based geopolymer mortar. The collected data contained various mix proportions and different curing ages (1 to 28 days), as well as different curing temperatures. The CS of all types of cementitious composites, including geopolymer mortars, is one of the most important properties; thus, developing a credible model for forecasting CS has become a priority. Therefore, in this study, three different models, namely, linear regression (LR), multinominal logistic regression (MLR), and nonlinear regression (NLR) were developed to predict the CS of geopolymer mortar. The proposed models were then evaluated using different statistical assessments, including the coefficient of determination (R^2^), root mean squared error (RMSE), scatter index (SI), objective function value (OBJ), and mean absolute error (MAE). It was found that the NLR model performed better than the LR and MLR models. For the NLR model, R^2^, RMSE, SI, and OBJ were 0.933, 4.294 MPa, 0.138, 4.209, respectively. The SI value of NLR was 44 and 41% lower than the LR and MLR models’ SI values, respectively. From the sensitivity analysis result, the most effective parameters for predicting CS of geopolymer mortar were the SiO_2_ percentage of the FA and the alkaline liquid-to-binder ratio of the mixture.

## 1. Introduction

To meet the demands of the construction industry, Portland cement production has increased significantly in recent years. However, the cement manufacturing industry is a significant contributor to rising carbon dioxide emissions into the atmosphere [1,2]. Because of this, it is now required for all countries to consider CO_2_ emission regulations and reductions [3]. As a result, there has been much research undertaken to find a new material that can be used instead of Portland cement. For example, geopolymer technology, which was first developed by Davidovits in France in 1970 [4].

Geopolymers are novel cementitious materials that have the potential to completely replace conventional Portland cement composites while emitting less CO_2_ [5]. The term ‘geopolymer’ refers to the binder product of an alkaline liquid reaction with silicon (Si) and aluminum (Al) in the source material [6]. Geopolymers are more cost-effective, sustainable, and resilient infrastructures because they are hard, stable at high temperatures, and cost-effective. They represent promising alternative binders with superior properties [7]. 

Geopolymer mortar is produced by mixing sand, an alkaline solution, and source binder materials. The alkaline solution consists of a mixture of sodium silicate and sodium hydroxide; sodium hydroxide with a purity of around 97 percent is used and comes in two major forms: pellets and flakes. Sodium silicate composition consists of three major compounds: SiO_2_, Na_2_O, and H_2_O [8]. The source binder materials include materials rich in SiO_2_ and Al_2_O_3_, such as fly ash. The source binder materials’ performance and reactivity are primarily determined by their chemical composition, fineness, and glassy phase content. Fly ash is the most common source binder material for making geopolymers because it is cheap, easy to find, and has more potential for making geopolymers. Fly ash is the finely chopped residue that is left over after coal is pulverized and burned. It is moved from the combustion chamber through electrostatic precipitators, or other devices that remove the particles before they reach the chimneys [9]. 

Fly ash is used as a self-cementitious material in geopolymer cement, completely replacing cement. The presence of aluminosilicate phases (from SiO_2_ and Al_2_O_3_) causes the structural bonds (-Si-O-Al-O-Si) to form through a polymerization mechanism [10]. When an alkaline solution comes into contact with a source binder material that has aluminosilicate in it, a chemical reaction called polymerization takes place. During the polymerization process, a polymeric chain and ring structure made up of Si-O-Al-O linkages is formed, with an empirical formula that looks like this: nM_2_O.Al_2_O_3_.xSiO_2_.yH_2_O; where M is an alkali action, n is the percentage polymerization, and w is the content of water. In addition, the geopolymerization reaction according to [11] can be described as follows.


**(Si_2_O_2_Al_2_O_2_)_n_+ H_2_O + OH^−^ → Si(OH)_4_+ Al(OH)^−4^**

**          → {-Si-O-Al-O-}_n_ + aH_2_O**



**2(Si_2_O_5_, Al_2_O_2_) + K_2_ (H_3_SiO_4_)_2_ + Ca(H_3_SiO_4_) → (8SiO_2_, _2_Al_2_O_3_, nH_2_O)**


Fauzi et al. [8] investigated the characterization of fly ash. They discovered that class F fly ash contained a higher concentration of Si and Al than class C fly ash. Chemical content affects the formation of (C-S-H) and (C-A-H) gels, as well as chemical bond gels (-Si-O-Al- and -Si-O-Si-). Unreacted FA particles are used as a filler in the cementitious process. According to the Australian Ash Development Association [12], the primary oxides in Australian fly ash are SiO_2_ and Al_2_O_3_, which account for around 80% of the total weight of the fly ash. The fly ash contains fewer than twenty percent by weight of components such as CaO and Fe_2_O_3_. The cumulative weight of oxides, such as Na_2_O, K_2_O, MgO, and SO_3_ in the fly ash, is less than 5% by weight. SiO_2_ is available in two microstructural forms (phases): crystal and amorphous. In an alkaline activator, amorphous SiO_2_ is more dissolvable (reactive) than crystal SiO_2_. The use of low-cost inert filler, binders, and recycled, as well as natural, aggregates for the production of geopolymers has been reviewed [13]; for instance, Kuenzel et al. [14] studied the effect of sand on the mechanical properties of metakaolin-based geopolymer. They concluded that adding sand to the geopolymer reduces the paste’s viscosity. The viscosity of the mix is reduced when the water content is reduced to achieve the same fraction of the solid volume. When 25% of the geopolymer pastes were replaced with sand, the mechanical properties of the samples improved, and it was observed that when the geopolymer specimens were heated to 110 °C, the mixtures containing at least 10% sand did not shrink; this is because sand forms a network that supports a fixed void volume. The compressive strength of the mortar was increased by increasing the volume percentage of aggregate [14].

Numerous alkaline activators are used to activate the fly ash in order to create geopolymer. In geopolymer mixing, sodium hydroxide (NaOH) or potassium hydroxide (KOH) is combined with sodium silicate (Na_2_SiO_3_) or calcium silicate (Ca_2_SiO_3_) for activation. The types and concentrations of the components used to prepare the alkaline activator dictate the activator’s properties (i.e., SiO_2_, H_2_O, Na_2_O). It was discovered that class F fly ash is an excellent source for geopolymer mortar and that sodium-based alkaline activators are more effective at activating fly ash than potassium-based activators [15]. The degree of polymerization between the binder materials and the alkaline solutions is affected by different circumstances. Soutsos et al. [16] studied the factors that affect the CS of geopolymers. They produced high strength geopolymer up to 70 MPa by using a mixture of NaOH and Na_2_SiO_3_ as an alkaline solution. The presence of silicates in the solution significantly influenced geopolymer strength. 

Various ratios of the alkaline liquid-to-binder ratio (l/b) have been considered to investigate their effect on geopolymer strength; for instance, Verma and Dev [17] studied the effect of the l/b ratio on the physical, mechanical, chemical, non-destructive test, and microstructural properties of concrete samples made with geopolymer. Their results showed that workability was increased with an increasing l/b ratio; however, the density of the concrete specimens decreased with increment in l/b ratios. Moreover, they reported that the CS, rebound number, and ultrasonic pulse velocity of the geopolymer concrete samples increased with increase in the l/b ratio from 0.4 to 0.6, and then decreased. The authors concluded that a maximum value for flexural and splitting tensile strength could be obtained at an l/b of 0.6. Similarly, Jeyasehar et al. [18] reported that the workability of geopolymers improved with increasing alkaline activator to fly ash ratio.

Chithambaram et al. [19] examined how the molarity of NaOH, curing regime, and use of GGBFS as a partial replacement for fly ash, changed the properties of geopolymer mortar. It was found that the CS of mortar specimens with 12 M of NaOH at 90 °C had the highest value. It was shown that mortar mixture strength increased as the replacement level of fly ash with GGBFS increased in an ambient curing condition; however, the product’s strength was decreased by nearly one-third that of geopolymer mortar made with fly ash as a precursor. Mishra et al. [20] investigated the effects of NaOH molarity on the strength of geopolymer concrete at various curing ages. They reported that, as the molarity of sodium hydroxide increased from 8 to 16 M, the tensile and compressive strengths of the geopolymer mortar specimens increased.

In most cementitious systems, curing conditions significantly impact mechanical strength development. In fly-ash-based geopolymer systems, a curing heat regime between 40 °C to 100 °C, for a duration of 4–48 h, inside ovens provided good conditions for geopolymer synthesis [21]. However, geopolymers have been successfully prepared at room temperature using calcined source binder materials of pure geological origin, such as metakaolin [22]. According to Hardjito et al. [23], a higher curing temperature does not always imply a higher compressive strength, especially above 60 °C.

Hardjito [22] investigated the effects of curing duration inside ovens; he cured geopolymer concrete samples at 60 °C for 4 to 96 h. His results revealed that curing time for a longer period enhanced the polymerization process and yielded a higher strength. Furthermore, it was concluded that curing of geopolymer specimens under low temperature for a longer curing time inside ovens did not weaken the strength of the samples; however, the sample strength was reduced after longer curing at high temperatures due to the breakdown of the synthesized geopolymer structure, resulting from excessive shrinkage of the geopolymer gel [24,25]. Vijai et al. [26] studied the effects of different curing regimes on the strength properties of geopolymer concrete, concluding that hot-cured specimens have a much higher CS than ambient-cured geopolymer concrete samples. They also reported that the CS increased as the age of the concrete increased from 7 to 28 days in ambient curing regimes, while this improvement in the CS was not significant in hot curing conditions. 

The effects of fly ash characteristics and alkaline activator components on the CS of fly-ash-based geopolymer mortar were investigated by Hadi et al. [27]. They prepared 180 different geopolymer mortar mixtures, then, an artificial neural network was successfully designed to develop models to forecast the CS of their mortar specimens. They used nine input parameters in their model, including fly ash and alkaline activator properties; however, some other important parameters, such as the molarity of NaOH, sand content, curing temperature, curing time, and age of the samples were absent from their input model parameters. 

Machine learning methods and empirical equations have been used to model various properties of concretes, such as the CS of green concrete [28], the CS of concrete containing a high volume of fly ash [29], the CS of fly-ash-based geopolymer concrete composites [30], and the CS of nano-silica modified self-compacting concrete [31]. For instance, Mohammed et al. [29] successfully used 450 datasets to create different models, including artificial neural networks, M5P-tree, linear regression, nonlinear regression, and multinominal logistic regression, to estimate the CS of high-volume fly ash concrete. Applying statistical assessment tools including R^2^, RMSE, and MAE it was found that the ANN, M5P-tree, and multi-logistic regression models could accurately predict the CS of large volume fly ash concrete. In addition, in previous papers [30,31], by using various model input parameters, we developed three different models to predict the CS of fly-ash-based geopolymer concrete and self-compacting concrete modified with nano-silica, respectively. We found that this issue had not been studied for geopolymer mortars; therefore, in this paper, we concentrate on different proposed models for the prediction of the CS of geopolymer mortars based on thirteen different model input parameters. The main difference between geopolymer concrete and mortars relates to their mix proportions and mixture constitutuents, while self-compacting concrete is totally different from geopolymer mortars. 

Many different parameters influence the CS of geopolymer mortars; however, there is a dearth of studies examining the effect of various mixture proportion parameters on the CS of fly-ash-based geopolymer mortars at varying curing temperatures and ages. As a result, the effects of thirteen different parameters on the CS of fly-ash-based geopolymer mortar were considered and quantified in this study using a variety of modeling techniques, including LR, NLR, and MLR models. Then various statistical tools, including RMSE, MAE, SI, OBJ, and R^2^, were used to assess the accuracy of the created models.

Finally, in this research, 247 datasets were collected from previous studies and analyzed to achieve the following goals:(a)Statistical analysis of the data to evaluate the effect of mixture compositions, such as fly ash content, sand, NaOH content, Na_2_SiO_3_ content, curing time and temperature, liquid-to-binder ratio, and the age of the CS of fly-ash based geopolymer.(b)Development of reliable models to predict the CS of FA-based geopolymer mortar. These models are useful for construction industries to produce a geopolymer mortar of high-quality and strength, while saving time and effort.(c)To ensure that the construction industry can use the models developed without the need for any experimental testing or theoretical constraints.(d)Exploration of the most influential parameters of FA-based geopolymer mortar for compressive strength.

## 2. Methods and Materials

### 2.1. Methodology

A total of 247 experimental datasets from past studies were collected and statistically analyzed. A summary of the FA-based geopolymer mortar is shown in Table 1, in which measured CS, and different mix quantities can also be seen. Based on the parameters used by various researchers, a database for FA-based geopolymer was created. The dataset was divided into three groups: training, testing, and validation. The training dataset contained 167 data features that were used to create the model; the testing and validation datasets each contained 40 data features. The latter two groups were used to evaluate the prediction models.

As can be seen in Table 1, the variables considered as input model parameters were FA (kg/m^3^), SiO_2_ and Al_2_O_3_ percent of the FA, sand content (kg/m^3^), NaOH (kg/m^3^), Na_2_SiO_3_ (kg/m^3^), SiO_2_/Na_2_O and H_2_O/Na_2_O of the sodium silicate, liquid-to-binder ratio (l/b), NaOH molarity (M), curing temperature, curing time, and age (days). Compressive strength was the only dependent variable. The collected datasets were used to estimate the CS of fly-ash-based geopolymer mortar using various models. Figure 1 illustrates the flowchart for the procedure used in this study. Subsequent sections present and discuss the details of the procedure, such as data collection, analysis, modeling, and evaluation.

### 2.2. Characteristics of Model Input Parameters 

A statistical analysis was carried out to determine whether there were any significant relationships between input parameters and the compressive strength of fly-ash-based geopolymer mortar. In this regard, all variables were taken into account, including (i) FA content, (ii) SiO_2_ (%), (iii) Al_2_O_3_(%), (iv) sand content (v) NaOH content, (vi) Na_2_SiO_3_ content, (vii) SiO_2_/Na_2_O of silicate solution, (viii) H_2_O/Na_2_O of silicate solution, (ix) l/b, (x) NaOH (M), (xi) curing temperature, (xii) curing time, and (xiii) age. Histograms were plotted to illustrate the underlying frequency distribution (shape) of the set of continuous material properties datasets. This enabled inspection of the data with respect to its underlying distribution (e.g., normal distribution), outliers, skewness, etc. The statistical analysis of the parameters is summarized in Table 2.

#### 2.2.1. Fly Ash Content (FA)

The types of fly ash used to prepare geopolymer mortars were classes F and C, according to the 247 datasets collected. The range of content was 460 to 909 kg/m^3^, with a median of 520 kg/m^3^. The standard deviation (SD) was 136.64 (kg/m^3^), the variance was 18,670.26 (kg/m^3^), and the skewness and kurtosis were 1.64 and 1.16, respectively. Figure 2 shows the FA/CS relationship; the histogram for FA content is also presented.

#### 2.2.2. SiO_2_ (FSO)

The range of SiO_2_ percentage content of the FA used in the geopolymer mortar was from 43 to 77 (%) depending on the type of fly ash used in geopolymer production. The SD, variance, kurtosis, and skewness were 10.25, 105.07, −0.69, and −0.02, respectively. The variation of SiO_2_ with CS and the histogram of SiO_2_ (%) are shown in Figure 3.

#### 2.2.3. Al_2_O_3_ (FAO)

Based on the collected datasets, the range of Al_2_O_3_ was found to be from 15 to 33 (%). This value depended on the type of fly ash used in the geopolymer production. The SD, variance, kurtosis, and skewness were 5.06, 25.62, 0.36, and −0.61, respectively. The variation between the CS and Al_2_O_3_ content, and the histogram of Al_2_O_3_ (%), are shown in Figure 4.

#### 2.2.4. Sand Content (S)

The sand used in the geopolymer mortars included natural and manufactured sand, with content ranging from 750 to 1840 kg/m^3^. The SD, variance, skewness, and kurtosis values were 141.88, 20,130.33, −0.98, 4.13, respectively. The histogram of sand content and the variation of sand content with CS are shown in Figure 5.

#### 2.2.5. NaOH Content (SH)

NaOH content of the mixes varied between 53 and 179 kg/m^3^. The SD, variance, skewness, and kurtosis values were 25.84, 667.89, 0.47, −0.11, respectively. The histogram of NaOH content and the relationship of NaOH content with CS can be seen in Figure 6.

#### 2.2.6. Na_2_SiO_3_ Content (SS)

Based on the data collected, the added Na_2_SiO_3_ content was in a range from 124 to 293 kg/m^3^. The SD, kurtosis, variance, and skewness values were 40.87, −0.84, 1670.7, −0.02, respectively. The variation of SS with the CS and the histogram of SS are shown in Figure 7.

#### 2.2.7. SiO_2_/Na_2_O of Silicate Solution (SO/N)

SiO_2_/Na_2_O ratio ranged from 2 to 3 with an SD of 0.3, variance of 0.09, kurtosis of 0.46, and skewness of 1.4. A histogram for SO/N was created, and its relationship with CS was represented, as shown in Figure 8.

#### 2.2.8. H_2_O/Na_2_O of Silicate Solution (H/N)

The range of the H/N ratio, based on the data collected, was 4 to 6. The SD, variance, skewness, and kurtosis were 0.64, 0.4, 1.6, and 1.6, respectively. A histogram was also created for H/N with respect to its relationship with the CS of fly-ash-based geopolymer mortars, as shown in Figure 9.

#### 2.2.9. Liquid-to-Binder Ratio (l/b)

The liquid-to-binder ratio varied from 0.4 to 0.8 based on the total data collected; additional water was included. The SD, variance, skewness, and kurtosis were 0.1, 0.01, 0.36, and −0.76, respectively. A histogram of l/b frequency and the variation of l/b with CS are shown in Figure 10.

#### 2.2.10. NaOH Molarity (M)

Based on the data collected from the literature, the variation of M with CS and a histogram for M were represented, as shown in Figure 11. After statistical analysis, it was found that the statistical parameters were as follows: the range of M was between 8 and 18 mol/L, the SD was 2.11 mol/L, the variance was 4.46, kurtosis was equal to −0.23, and skewness was equal to −0.39. 

#### 2.2.11. Curing Temperature (te)

When producing geopolymers, samples need to be heated for a period of time at elevated temperatures because increased temperature speeds up the polymerization process. Based on the collected datasets, the curing temperatures ranged from 25 to 80 °C, the SD was equal to 6.85 °C, the variance was 46.96 °C, skewness was −5.24, and the kurtosis was 29.47. A histogram and the relationship of curing temperature with CS are shown in Figure 12.

#### 2.2.12. Curing Time (t)

Another essential parameter that affects the CS of geopolymer composites is the curing period of geopolymer specimens inside ovens. Based on 247 datasets collected from the literature, it was found that geopolymer specimens were rested inside ovens for between 18 and 24 h, with an SD of 1.13, variance of 1.27, skewness of −4.98, and kurtosis of 22.97. A histogram of curing time and the relationship between CS and curing time are shown in Figure 13.

#### 2.2.13. Age (Ag)

The age of the tested samples from the collected datasets ranged from 1 to 28 days, with an SD and variance of 5.55 and 30.81, respectively, with kurtosis of 6.76 and skewness of 2.69. The relationship between age and CS and a histogram of age are shown in Figure 14.

#### 2.2.14. Compressive Strength

Based on the collected data from the literature, summarized in Table 1, the compressive strength ranged from 2 to 80 MPa, with an SD of 18.36 MPa, and variance, kurtosis, and skewness values of 337.01, −0.61, and 0.46, respectively. A histogram for CS of FA-based geopolymer mortar is shown in Figure 15.

### 2.3. Modeling

Based on the R^2^ value of Figure 2, Figure 3, Figure 4, Figure 5, Figure 6, Figure 7, Figure 8, Figure 9, Figure 10, Figure 11, Figure 12, Figure 13 and Figure 14, there was no direct relationship between the CS of fly-ash-based geopolymer mortar and thirteen input model parameters. As a result, three different models were developed to show the effects of the various mixture proportions referred to above on the CS of FA-based geopolymer mortar. The models proposed in this study were used to estimate the CS of FA-based geopolymer mortar with a view to selecting one providing the best estimate of CS with respect to measured CS based on experimental data. A number of criteria were considered when comparing the forecasts of the different models. These criteria were used to find a scientifically valid model, based on the least difference between the measured data and the data that was predicted using the model. In addition, the proposed models needed to have low RMSE, OBJ, and SI values, and high R^2^ values. 

#### 2.3.1. Linear Regression (LR) Model

LR is a well-established method for estimating and forecasting the CS of concrete composites [30]. This model has a general form, which is illustrated in Equation (1)
(1)$$\sigma_{c}=Y_{o}+A\left(\frac{w}{b}\right)$$

Y_o_ and A, σ_c_, and w/c denote the equation parameters, compressive strength, and water to cement ratio. Here, other parameters, such as curing time and other ingredients, are not included. Therefore, another formula was used considering all parameters, as suggested by Faraj et al. [31], and represented by Equation (2)
(2)$$\sigma_{c}=Y_{o}+a\left(\frac{w}{b}\right)+b\left(t\right)+c\left(\mathrm{FA}\right)+d\left(\mathrm{CA}\right)+e\left(\mathrm{SP}\right)+f\left(\mathrm{NS}\right)+g\left(C\right)$$

In this equation, NS represents the nano-silica content (percent), C represents the cement content (kg/m^3^), w/b represents the water-to-binder ratio (percent), t represents the curing time (days), SP represents the superplasticizer content (kg/m^3^), FA represents the fine aggregate content (kg/m^3^). CA represents the coarse aggregate content (kg/m^3^). In addition, the model parameters are denoted by the letters a, b, c, d, e, f, g, and h. The proposed Equation (2) can be viewed as an extension of Equation (1) because all variables can be adjusted linearly; however, while all variables can influence CS and interact with one another, this is not always the case. As a result, in order to estimate the CS accurately, the model should always be adapted [36]. In this study, the above equation was used to predict the CS of FA-based geopolymer mortar with parameters considered for geopolymer mortar included, modifying Equation (2) to produce Equation (3).
(3)$$\begin{array}{l} \sigma_{c}=Y_{o}+a\left(\mathrm{FA}\right)+b\left(\mathrm{FSO}\right)+c\left(\mathrm{FAO}\right)+d\left(S\right)+e\left(\mathrm{SH}\right)+f\left(\mathrm{SS}\right)+\\ g\left(\frac{\mathrm{SO}}{N}\right)+h\left(\frac{H}{N}\right)+i\left(\frac{l}{b}\right)+j\left(M\right)+k\left(\mathrm{te}\right)+l\left(t\right)+m\left(\mathrm{Ag}\right)\ldots \ldots .. \end{array}$$
where FA, FSO, FAO, S, SH, SS, SO/N, SSH/N, l/b, M, Cte, Ct, and t are fly ash content (kg/m^3^), SiO_2_ (%) of the fly ash, Al_2_O_3_ (%) of the fly ash, sand content (kg/m^3^), NaoH content (kg/m^3^), Na_2_SiO_3_ content (kg/m^3^), SiO_2_/Na_2_O of the silicate solution, H_2_O/Na_2_O of the silicate solution, liquid-to-binder ratio, NaOH molarity, curing temperature, curing time, and age of concrete specimens, respectively, whereas, Y_o_, a, b, c, d, e, f, g, h, I, j, k, l, and m are model parameters.

#### 2.3.2. Multinominal Logistic Regression (MLR) Model

Multinominal logistic regression (MLR) is a regression algorithm conducted if the predicate variable has a nominal value above two stages. In other words, MLR is a statistical technique that is similar to multiple linear regression, but can be used to clarify the difference between a nominal predictor variable and one or more independent variables [29], as shown in Equation (4).
(4)$$\sigma_{c}=a{\left(\mathrm{FA}\right)}^{b}{\left(\mathrm{FSO}\right)}^{c}{\left(\mathrm{FAO}\right)}^{d}{\left(S\right)}^{e}{\left(\mathrm{SH}\right)}^{f}{\left(\mathrm{SS}\right)}^{g}{\left(\frac{\mathrm{SO}}{N}\right)}^{h}{\left(\mathrm{HN}\right)}^{i}{\left(\frac{l}{b}\right)}^{j}{\left(M\right)}^{k}{\left(\mathrm{te}\right)}^{l}{\left(t\right)}^{m}{\left(\mathrm{Ag}\right)}^{n}$$
where FA, FSO, FAO, S, SH, SS, SO/N, SSH/N, l/b, M, Cte, Ct, and t are fly ash content (kg/m^3^), SiO_2_ (%) of the fly ash, Al_2_O_3_ (%) of the fly ash, sand content (kg/m^3^), NaoH content (kg/m^3^), Na_2_SiO_3_ content (kg/m^3^), SiO_2_/Na_2_O of the silicate solution, H_2_O/Na_2_O of the silicate solution, liquid-to-binder ratio, NaOH molarity, curing temperature, curing time, and age of concrete specimens, respectively, whereas, Y_o_, a, b, c, d, e, f, g, h, I, j, k, l, m, and n are model parameters.

#### 2.3.3. Nonlinear Regression (NLR) Model

Equation (5) can be applied as a common form to develop a nonlinear regression model [30,31]
(5)$$\begin{array}{l} \sigma_{c}={}^{\beta_{1}}{\left(\mathrm{FA}\right)}^{\beta_{2}}{\left(\mathrm{FSO}\right)}^{\beta_{3}}{\left(\mathrm{FAO}\right)}^{\beta_{4}}{\left(S\right)}^{\beta_{5}}{\left(\mathrm{SH}\right)}^{\beta_{6}}{\left(\mathrm{SS}\right)}^{\beta_{7}}{\left(\frac{\mathrm{SO}}{N}\right)}^{\beta_{8}}{\left(\frac{H}{N}\right)}^{\beta_{9}}{\left(\frac{l}{b}\right)}^{\beta_{10}}{\left(M\right)}^{\beta_{11}}{\left(\mathrm{Cte}\right)}^{\beta_{12}}{\left(\mathrm{Ct}\right)}^{\beta_{13}}{\left(t\right)}^{\beta_{14}}\\ +{}^{\beta_{15}}{\left(\mathrm{FA}\right)}^{\beta_{16}}{\left(\mathrm{FAO}\right)}^{\beta_{17}}{\left(S\right)}^{\beta_{18}}{\left(\mathrm{SH}\right)}^{\beta_{19}}{\left(\mathrm{SS}\right)}^{\beta_{20}}{\left(\frac{\mathrm{SO}}{N}\right)}^{\beta_{21}}{\left(\frac{H}{N}\right)}^{\beta_{22}}{\left(\frac{l}{b}\right)}^{\beta_{23}}{\left(M\right)}^{\beta_{24}}{\left(\mathrm{te}\right)}^{\beta_{25}}{\left(t\right)}^{\beta_{26}}{\left(\mathrm{Ag}\right)}^{\beta_{27}} \end{array}$$
where FA, FSO, FAO, S, SH, SS, SO/N, SSH/N, l/b, M, Cte, Ct, and t are fly ash content (kg/m^3^), SiO_2_ (%) of the fly ash, Al_2_O_3_ (%) of the fly ash, sand content (kg/m^3^), NaOH content (kg/m^3^), Na_2_SiO_3_ content (kg/m^3^), SiO_2_/Na_2_O of the silicate solution, H_2_O/Na_2_O of the silicate solution, liquid-to-binder ratio, NaOH molarity, curing temperature, curing time, and age of concrete specimens, respectively, while, β_1_, β_2_, β_3_, ……. β_27_ are model parameters.

## 3. Model Evaluation Criteria

Numerous performance parameters, including the coefficient of determination (R^2^), root mean squared error (RMSE), mean absolute error (MAE), the scatter index (SI), the OBJ, T_stat_, and U_95_, were used to evaluate and assess the proposed models’ efficiency:(6)$$R^{2}=1-\frac{\sum_{1}^{n}{\left(\mathrm{yp}-\mathrm{ye}\right)}^{2}}{\sum_{1}^{n}{\left(\mathrm{ye}-\bar{\mathrm{ye}}\right)}^{2}}$$
(7)$$\mathrm{RMSE}=\sqrt{\frac{\mathrm{SSE}}{n}}$$
(8)$$\mathrm{MAE}=\frac{\sum_{1}^{n}\left|\mathrm{yp}-\mathrm{ye}\right|}{n}$$
(9)$$\mathrm{MBE}=\frac{\sum_{1}^{n}\left(\mathrm{yp}-\mathrm{ye}\right)}{n}$$
(10)$$\mathrm{SI}=\frac{\mathrm{RMSE}}{\bar{\mathrm{ye}}}$$
(11)$$\mathrm{OBJ}=\left(\frac{n_{\mathrm{tr}}}{n_{\mathrm{to}}}*\frac{{\mathrm{RMSE}}_{\mathrm{tr}}+{\mathrm{MAE}}_{\mathrm{tr}}}{R^{2}{}_{\mathrm{tr}}+1}\right)+\left(\frac{n_{\mathrm{te}}}{n_{\mathrm{to}}}*\frac{{\mathrm{RMSE}}_{\mathrm{te}}+{\mathrm{MAE}}_{\mathrm{te}}}{R^{2}{}_{\mathrm{te}}+1}\right)+\left(\frac{n_{\mathrm{val}}}{n_{\mathrm{to}}}*\frac{{\mathrm{RMSE}}_{\mathrm{val}}+{\mathrm{MAE}}_{\mathrm{val}}}{R^{2}{}_{\mathrm{val}}+1}\right)$$
(12)$$T_{\mathrm{stat}}=\sqrt{\frac{\left(n-1\right){\mathrm{MBE}}^{2}}{{\mathrm{RMSE}}^{2}}}$$
(13)$$U_{95}=1.96*\sqrt{{\mathrm{SD}}^{2}+{\mathrm{RMSE}}^{2}}$$
where ye and yp are the experimental and the predicted values of the path pattern, correspondingly, and $\bar{\mathrm{ye}}$ is the average of the measured values. tr, te, and val are referred to as training, testing, and validation datasets, respectively, and n is the number of data items each dataset involves. 

The models were evaluated considering the criteria mentioned above, based on SI; each model was assessed as shown in Figure 16, which is adapted from [37]. For the R^2^ value, the best value is one; for the other statistical parameters, the lower the value, the better the model would be. The RMSE values of the models were compared; a lower value of RMSE and higher R^2^ jointly indicate better prediction amongst the models. 

## 4. Results

### 4.1. Relationship between Measured and Predicted CS Values

#### 4.1.1. The LR Model

The experimental and predicted CS variation for all training, testing, and validation data is shown in Figure 17a,b; the model parameters were determined using the least squares method. In this model, Equation (14) relates the weight of each model parameter to the output. This is how this model works. In Excel, Solver was used to determine the ideal value for an equation in a cell called the “objective cell,” which was used to determine how important each parameter was to the CS of fly-ash-based geopolymer mortar mixtures. The sum of error squares and the least-squares method was used to determine how important each parameter was to the CS.
(14)$$\begin{array}{c} \sigma_{c}=244.3-0.15\mathrm{FA}-1.43\mathrm{FSO}-0.56\mathrm{FAO}-0.085S+0.041\mathrm{SH}+0.079\mathrm{SS}\\ \;\;\;\;\;+66.88\left(\frac{\mathrm{SO}}{N}\right)-27.7\left(\frac{H}{N}\right)-30.6\left(\frac{l}{b}\right)+0.984M+0.212\mathrm{te}\\ \;\;\;\;\;+0.735t+0.428\mathrm{Ag} \end{array}$$

As can be seen from the model parameters, the SO/N, l/b, and H/N significantly affect the CS of FA-based geopolymer mortar. The evaluation parameters, including R^2^, RMSE, and MAE were 0.808, 7.305 MPa, and 5.794 MPa, respectively. SI and OBJ for the training dataset were 0.234 and 7.527, respectively. 

#### 4.1.2. The MLR Model

Figure 18a,b show the correlations between the actual and predicted CS of fly-ash-based geopolymer mortar mixtures for training, testing, and validation datasets. the model parameters were determined using the least squares method. Similar to the LR model, as illustrated in Equation (15), the weighting of model parameters revealed that l/b, H/N, and FA were the input variable parameters that significantly affected the CS of fly-ash-based geopolymer mortar mixtures.
(15)$$\begin{array}{l} \sigma_{c}=32.32{\mathrm{FA}}^{-5.85}{\mathrm{FSO}}^{-1.9}{\mathrm{FAO}}^{0.283}S^{-1.8}{\mathrm{SH}}^{1.145}{\mathrm{SS}}^{2.107}\\ \;\;\;\;{\left(\frac{\mathrm{SO}}{N}\right)}^{-3.56}{\left(\frac{H}{N}\right)}^{6.952}{\left(\frac{l}{b}\right)}^{-3.3}M^{0.17}{\mathrm{te}}^{0.33}t^{9.198}{\mathrm{Ag}}^{0.181} \end{array}$$

The evaluation parameters, including R^2^, RMSE, and MAE were 0.833, 6.83 MPa, and 5.2 MPa, respectively. SI and OBJ for the training dataset were 0.219 and 6.925, respectively. 

#### 4.1.3. The NLR Model

The predicted and observed CS of the fly-ash-based geopolymer mortar mixtures for the datasets are shown in Figure 19a,b. The model parameters were determined using the least squares method; the equation for the NLR model can be written as shown in Equation (16). In the MLR model, the most significant independent variables affecting the CS of fly-ash-based geopolymer mortar mixtures were FA content, H/N, l/b, and SO/N.
(16)$$\begin{array}{c} \sigma_{c}=53.4{\mathrm{FA}}^{-7.2}{\mathrm{FAO}}^{0.14}S^{-5}{\mathrm{SH}}^{0.38}{\mathrm{SS}}^{2.13}{\left(\frac{\mathrm{SO}}{N}\right)}^{-7}{\left(\frac{H}{N}\right)}^{11.8}{\left(\frac{l}{b}\right)}^{-2.1}\;M{\;}^{0.96}{\mathrm{te}}^{0.68}t^{15.4}{\mathrm{Ag}}^{0.24}\\ \;+7.69{\mathrm{FA}}^{-1.1}{\mathrm{FSO}}^{-14}{\mathrm{FAO}}^{9.42}S^{-1.7}{\mathrm{SH}}^{3.38}{\mathrm{SS}}^{4.01}{\left(\frac{\mathrm{SO}}{N}\right)}^{5.23}{\left(\frac{H}{N}\right)}^{-2.47}{\left(\frac{l}{b}\right)}^{-7.99}\;M^{1.12}{\mathrm{te}}^{-0.29}t^{2.07}{\mathrm{Ag}}^{0.1} \end{array}$$

R^2^, RMSE, and MAE were 0.934, 4.294 MPa, and 3.15 MPa, respectively. SI and OBJ for the training dataset were 0.138 and 4.209, respectively. 

### 4.2. Evaluation of Developed Model

As mentioned in Section 3, different statistical assessment tools, including R^2^, RMSE, MAE, SI, OBJ, T_stat_, and U_95_ were used to assess the developed models. The NLR was the best model among the developed models, having a higher R^2^ and lower RMSE and MAE values. The developed model comparison is shown in Figure 20 using the testing dataset. The residual errors for all models for the training, testing, and validation datasets are shown in Figure 21. From the two Figure 20 and Figure 21, the superior performance of the NLR model is clear with predicted compressive values close to actual values.

The SI values for developed models for all training, testing, and validation datasets are shown in Figure 22. As can be seen, from the figure, the SI values for NLR were 0.138, 0.172, and 0.129 for the training, testing, and validation datasets, respectively. The SI value was between 0.1 to 0.2 for the NLR model, indicating good performance; for the other two models, MLR and LR, the SI values ranged from 0.2 to 0.3, which indicates fair performance of these models. The SI value for NLR was 56% of the LR model’s SI value, and 59% of the MLR model’s SI value. Moreover, the MLR model was better than the LR model since it had lower SI, RMSE, MAE, and OBJ values and a higher R^2^ value.

The OBJ values of all models are shown in Figure 23; the figure shows the better performance of the NLR model with an OBJ value of 4.209, while the OBJ values for the LR and MLR models were 7.52 and 6.925, respectively.

The uncertainty and T-stat values are shown in Figure 24; for the testing dataset, the NLR model had higher U95 and T-stat values than the other two models. However, all developed models’ U95 and t-stat values were nearly identical.

### 4.3. Sensitivity Analysis

Sensitivity analysis was used to determine the most effective parameters for predicting the CS of fly-ash-based geopolymer mortar specimens; for this purpose, the MLR model was selected. Each time an input parameter was removed from the equation and R^2^, RMSE, MAE were recorded, the trail with minimum R^2^ and maximum RMSE and MAE values was selected. The removed parameter in that trial was the most effective parameter on CS, as shown in Table 3; bold means that FSO in row three is the most influential parameter in the prediction of the CS of fly-ash-based geopolymer mortar composites.

## 5. Conclusions

To develop reliable models for predicting the compressive strength of geopolymer mortar, a total of 247 datasets were collected from various research articles with varying mix proportions; after statistically analyzing the data and proposing models for CS prediction, the following conclusions were drawn:Class F fly ash can be used as an aluminosilicate source material to synthesize geopolymer mortar with a strength as high as 80 MPa at 28 days.The precursor (FA) characterization had a higher impact on the compressive strength of the geopolymer mortar; this was confirmed by sensitivity analysis.This study developed LR, MLR, and NLR models to predict CS of fly-ash-based geopolymer mortar; the NLR model was the best model amongst the proposed models with R^2^, RMSE, SI, and OBJ of 0.934, 4.29 MPa, 0.138, and 4.21 MPa, respectivelyThe SI value of the NLR model for all three phases (training, testing, and validation) was between 0.1 and 0.2, indicating good performance of the NLR model, the SI value being 44 and 41% lower than the LR and MLR model’s SI values, respectively.The OBJ value of the NLR model was 44 and 39% lower than the OBJ values for the LR and MLR models, respectivelySeveral evaluation criteria, including the root mean square error (RMSE), the coefficient of determination (R^2^), the OBJ, the SI, and the mean absolute error (MAE) were used to assess the models. NLR was the best model as determined in this study based on data acquired from the literature, showing a higher R^2^ value and lower MAE and RMSE values.The most effective parameters for prediction of the compressive strength of fly-ash-based geopolymer mortar were SiO_2_ (%) of the fly ash (FSO), Fa content, and then the l/b ratio.The determined sequence for optimal suitability and greater performance of the proposed models was NLR > LR > MLR.

## Figures and Tables

**Figure 1 materials-15-01868-f001:**
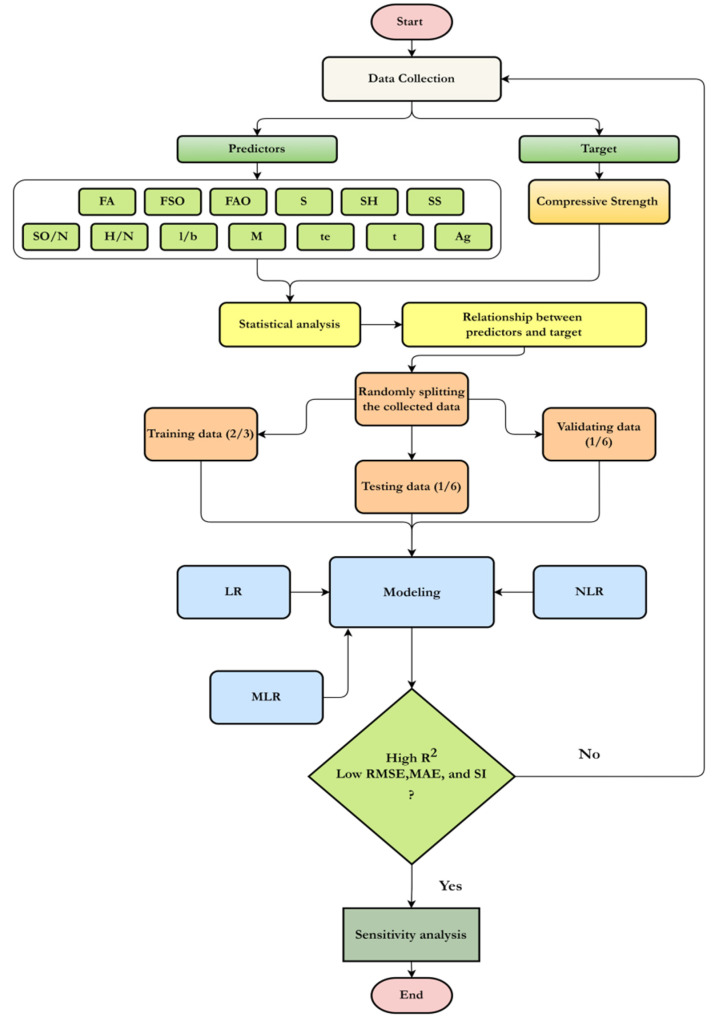
Flowchart diagram of the current study.

**Figure 2 materials-15-01868-f002:**
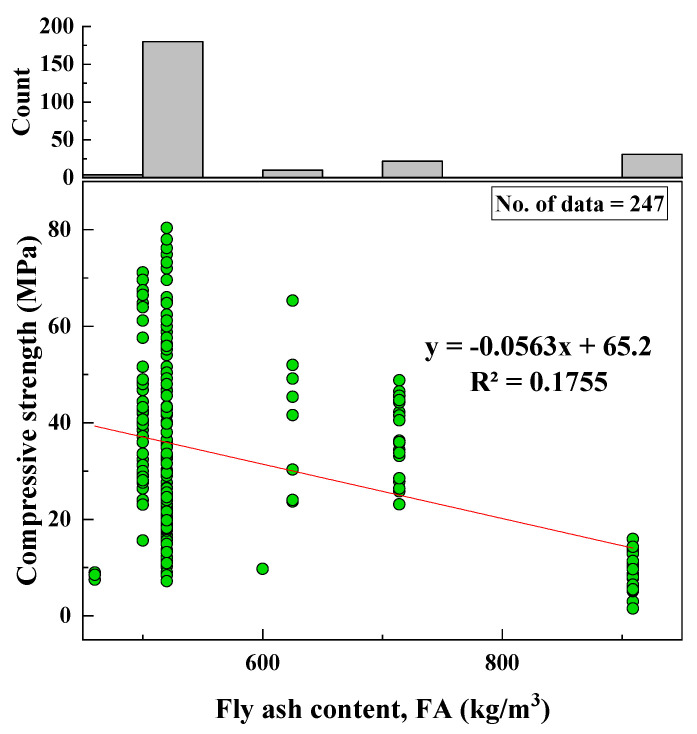
Correlation between CS and FA content and histogram of FA content.

**Figure 3 materials-15-01868-f003:**
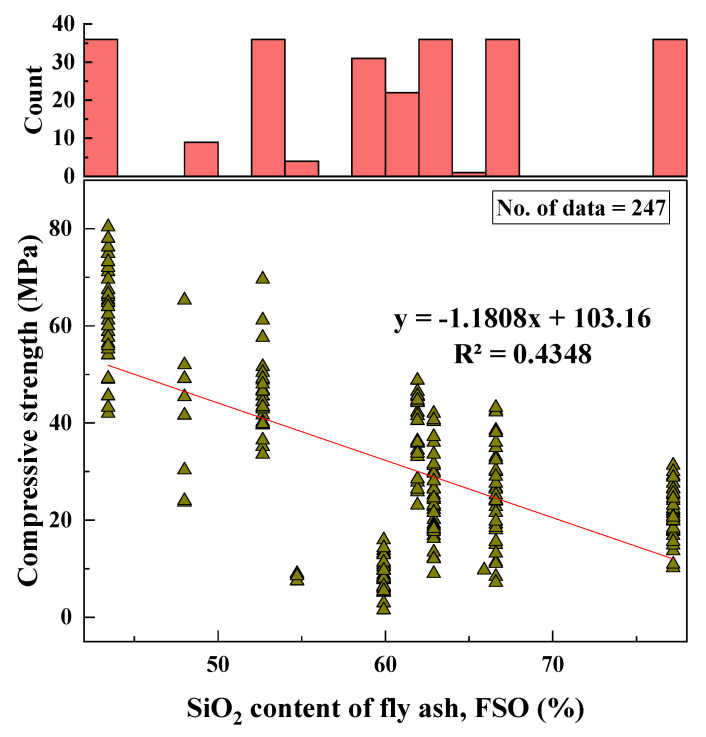
Variation of CS with FSO content and histogram of FSO.

**Figure 4 materials-15-01868-f004:**
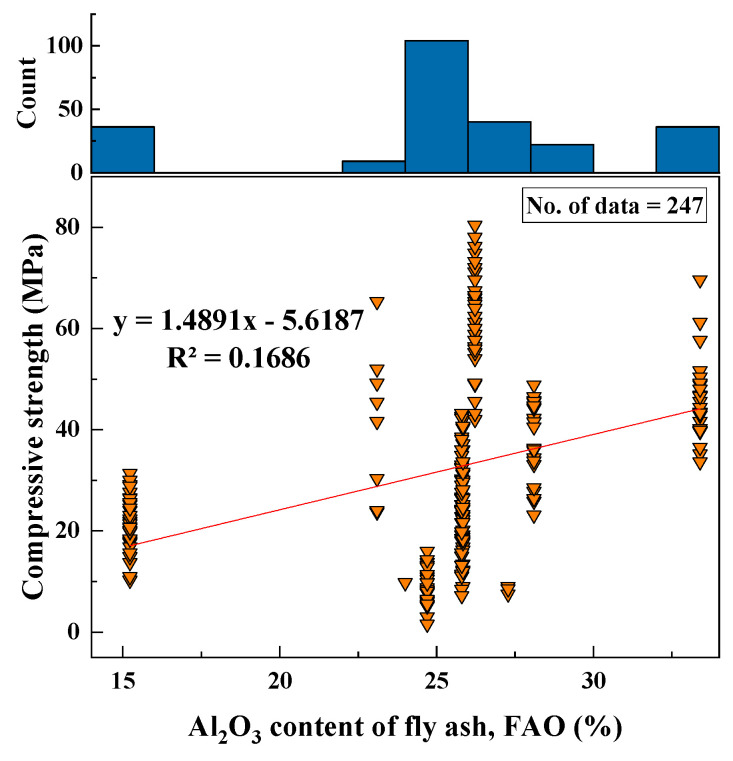
Relationship between CS and FAO content of FA and histogram of FAO.

**Figure 5 materials-15-01868-f005:**
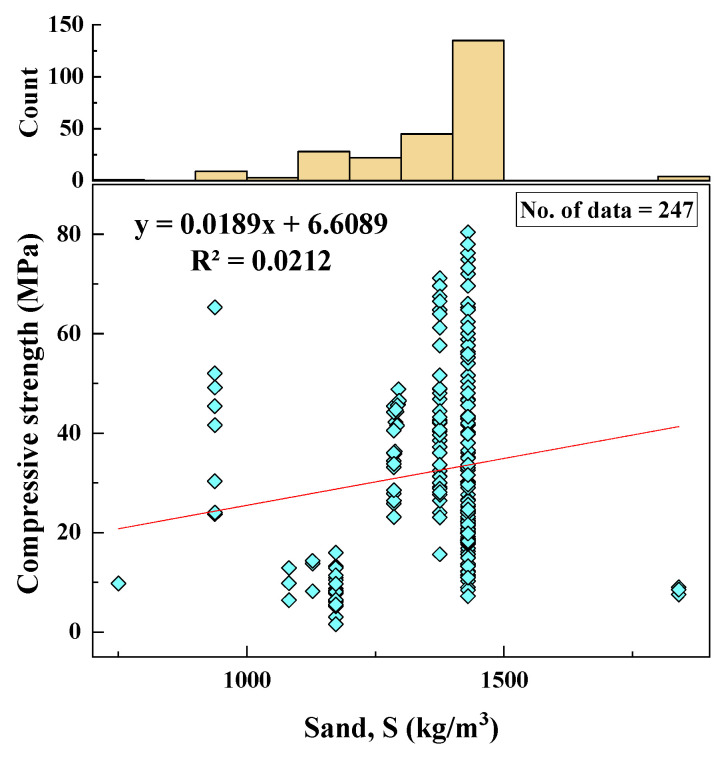
Correlation between CS and S content and histogram of S content.

**Figure 6 materials-15-01868-f006:**
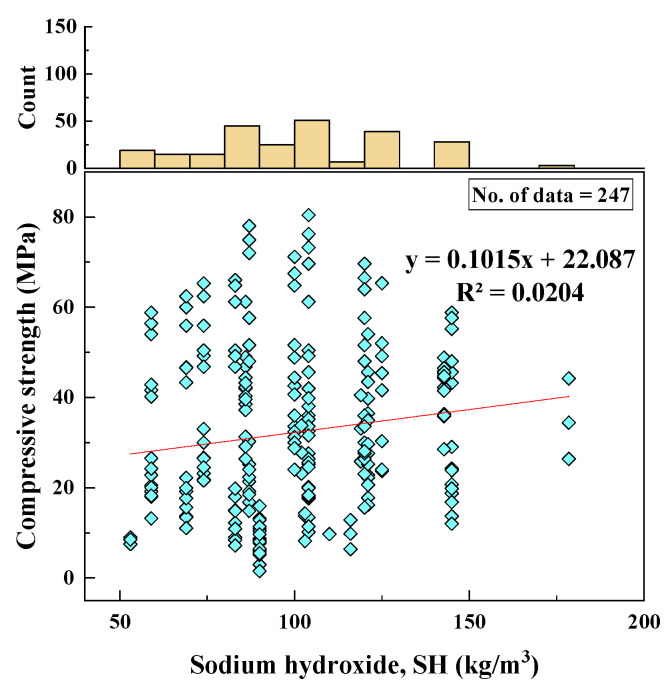
Relationship between CS and SH content and histogram of SH content.

**Figure 7 materials-15-01868-f007:**
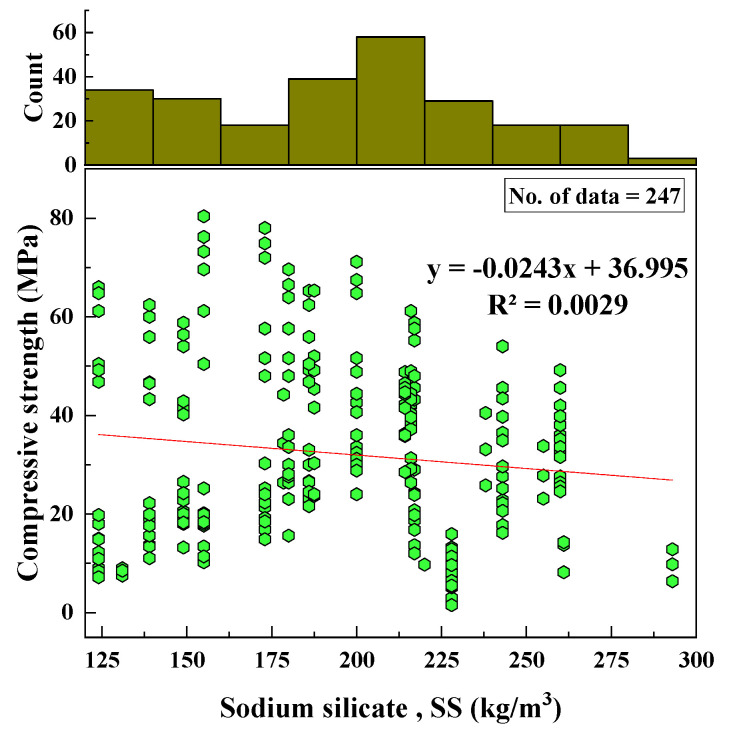
Relationship between CS and SS content and histogram of SS content.

**Figure 8 materials-15-01868-f008:**
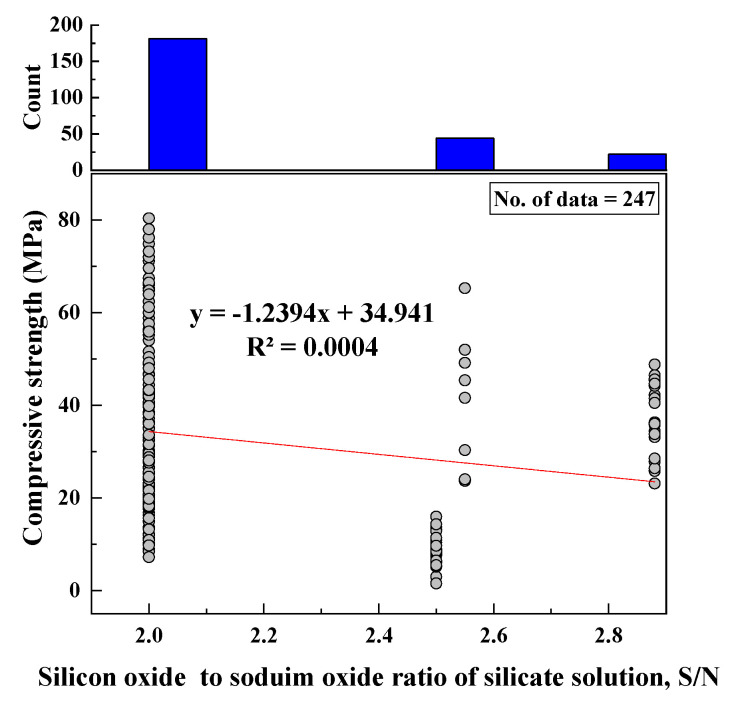
Correlations between CS and SO/N and histogram of SO/N.

**Figure 9 materials-15-01868-f009:**
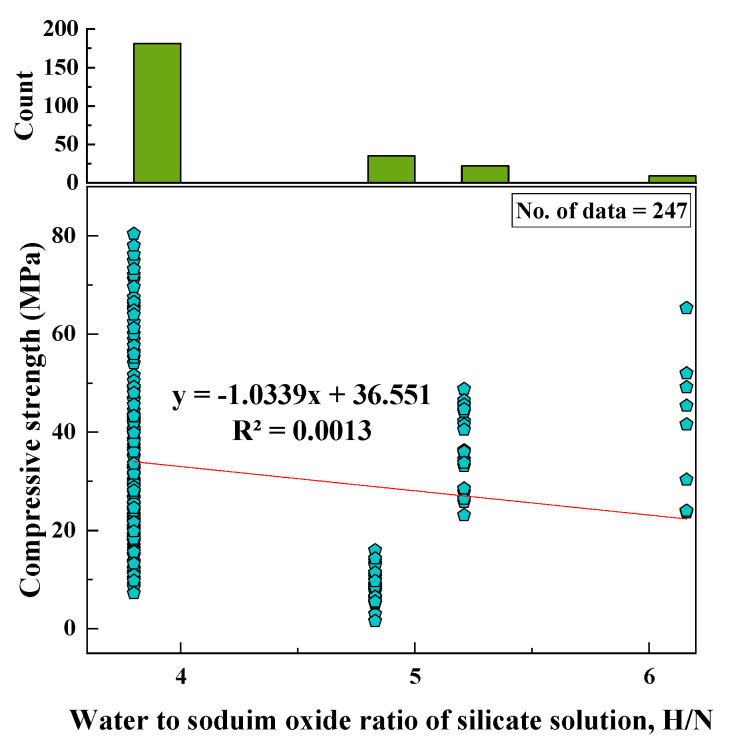
Relationship between CS and H/N and histogram of H/N.

**Figure 10 materials-15-01868-f010:**
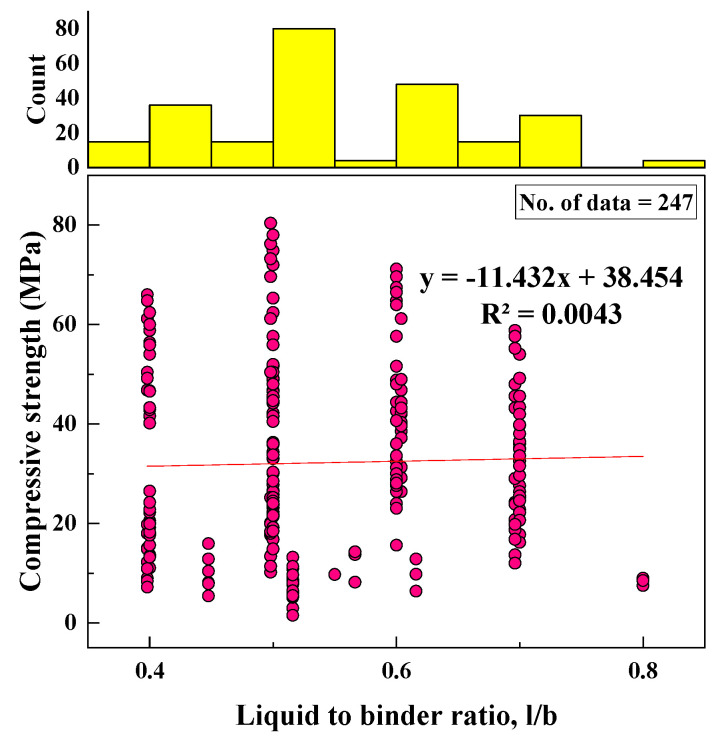
Variation between CS and l/b and histogram of l/b.

**Figure 11 materials-15-01868-f011:**
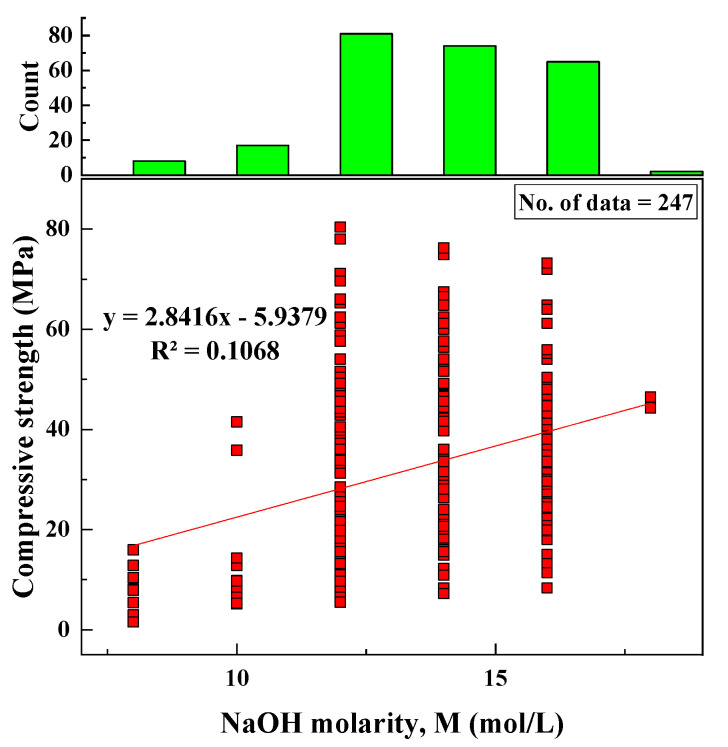
Variation between CS and M and histogram of M.

**Figure 12 materials-15-01868-f012:**
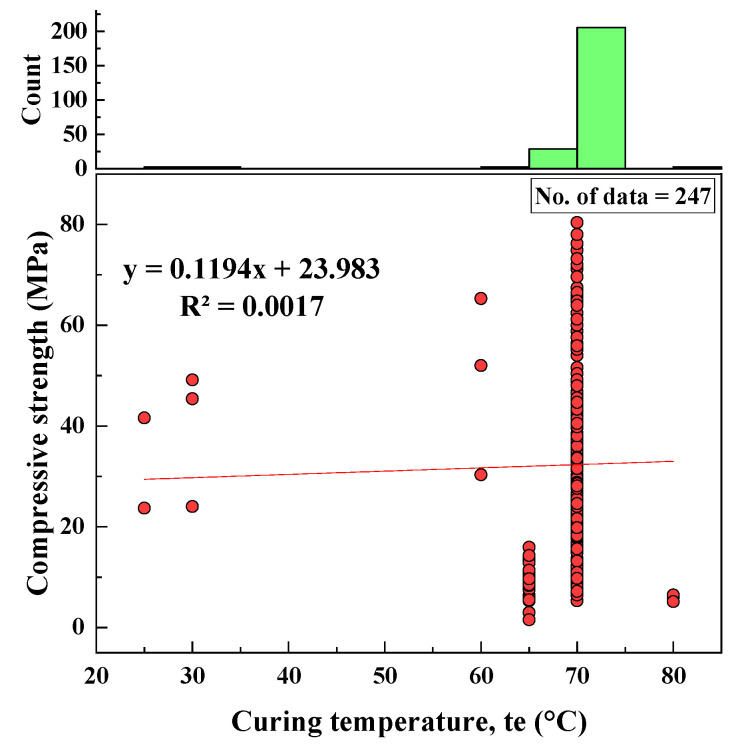
Correlation between CS and te and histogram of te.

**Figure 13 materials-15-01868-f013:**
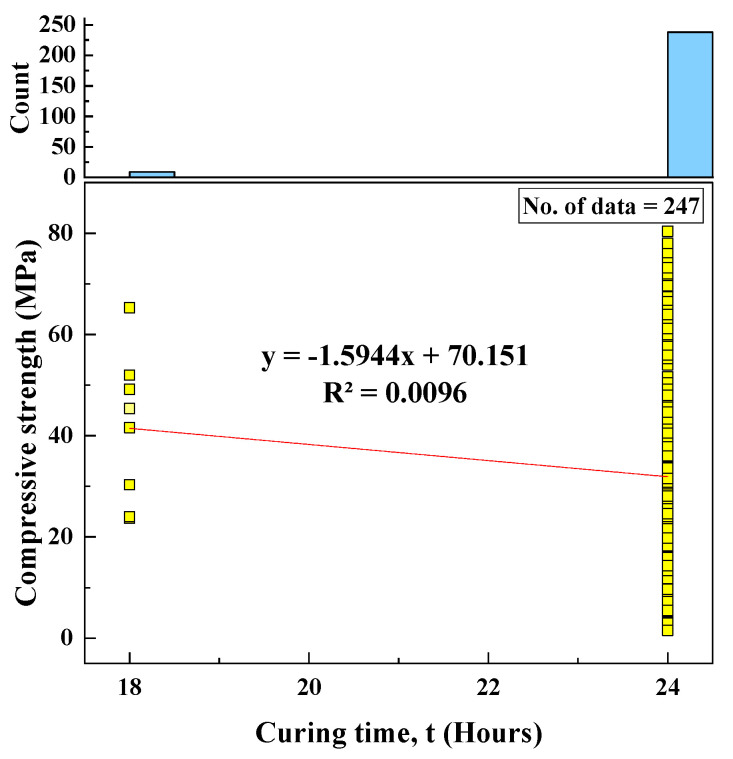
Correlation between the CS and curing time and histogram of curing time.

**Figure 14 materials-15-01868-f014:**
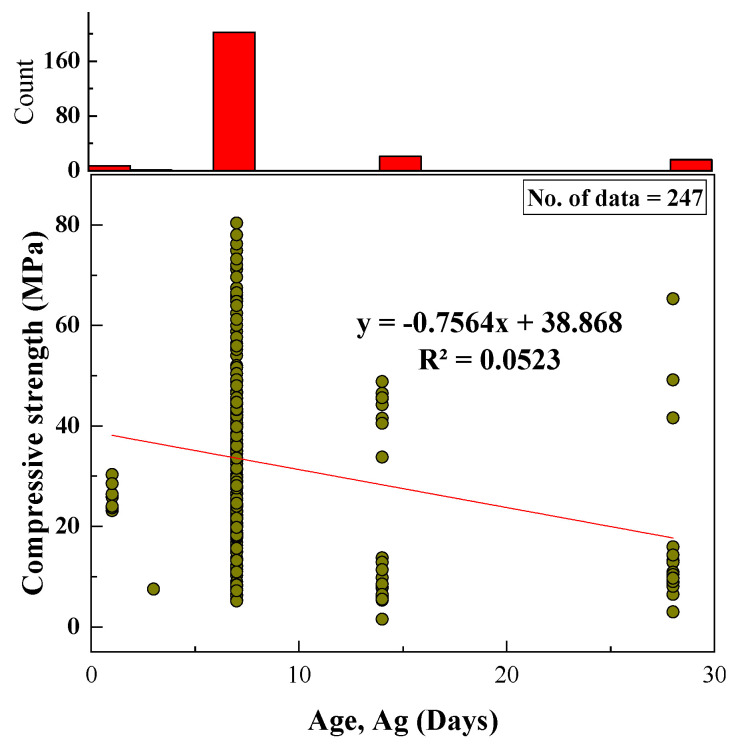
Relationship between CS and age and histogram of age of specimens.

**Figure 15 materials-15-01868-f015:**
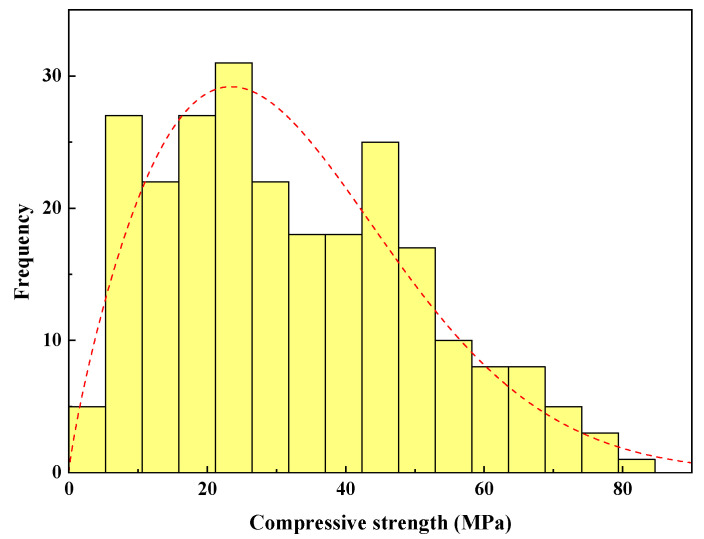
Weibull distribution function and histogram for CS of FA-based geopolymer mortar up to 28 days.

**Figure 16 materials-15-01868-f016:**
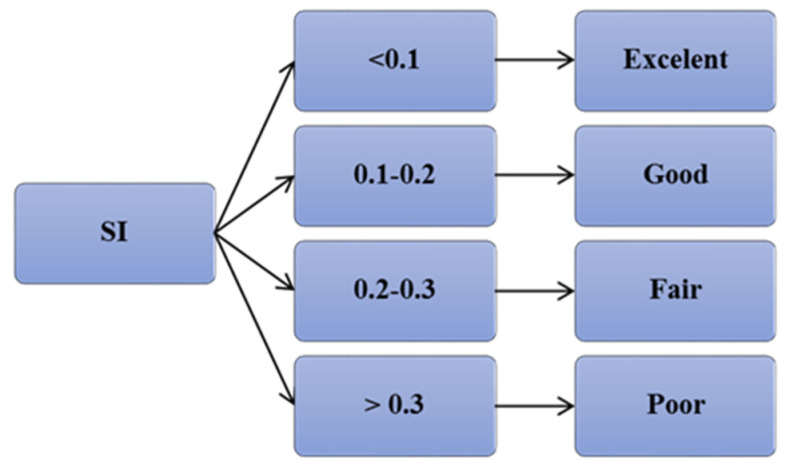
Performance of the models considering SI parameter.

**Figure 17 materials-15-01868-f017:**
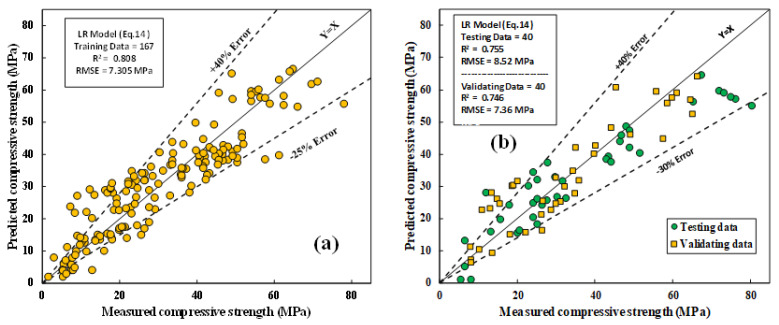
Correlations between tested and forecasted CS of FA-based geopolymer mortar using LR model; (**a**) training data; (**b**) testing and validation data.

**Figure 18 materials-15-01868-f018:**
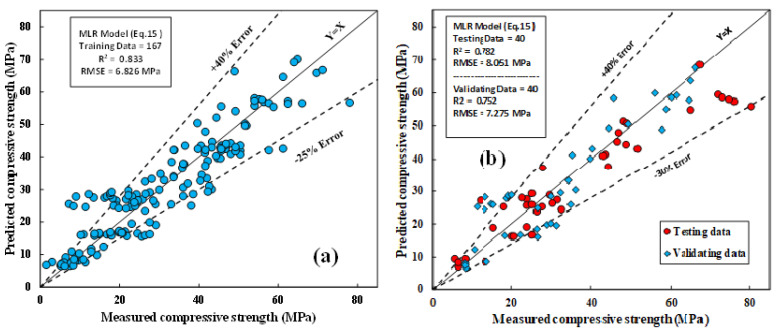
Correlations between tested and forecasted CS of FA-based geopolymer mortar using MLR model; (**a**) training data; (**b**) testing and validation data.

**Figure 19 materials-15-01868-f019:**
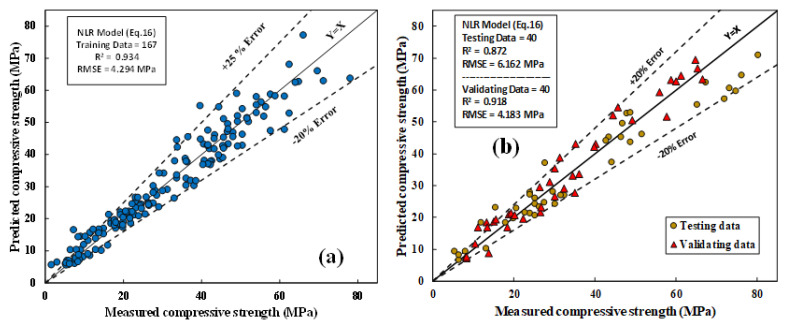
Correlations between tested and forecasted CS of FA-based geopolymer mortar using NLR model; (**a**) training data; (**b**) testing and validation data.

**Figure 20 materials-15-01868-f020:**
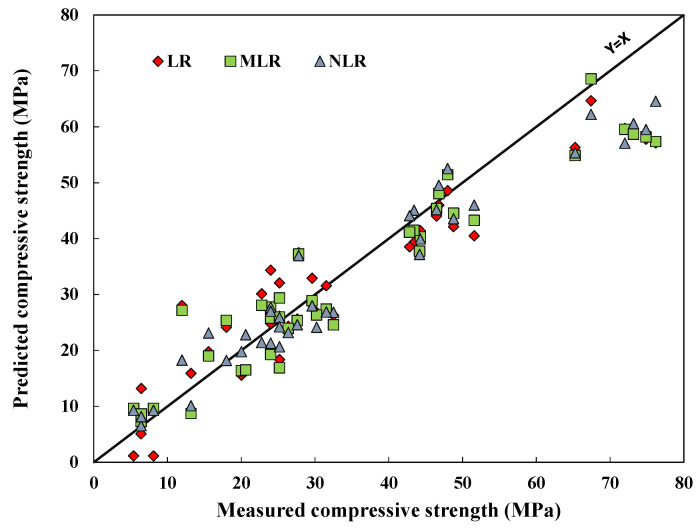
Correlations between the model forecasts of the CS of fly-ash-based geopolymer mortars using testing data.

**Figure 21 materials-15-01868-f021:**
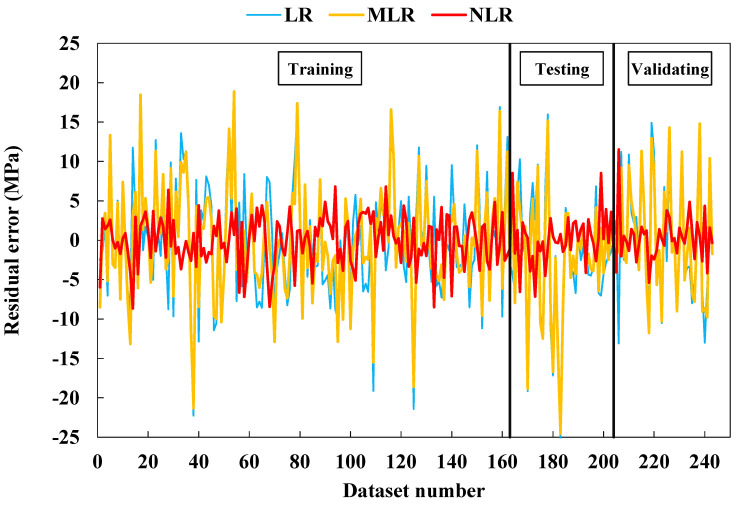
Variation in forecasted values of CS of fly-ash-based geopolymer mortars for all datasets.

**Figure 22 materials-15-01868-f022:**
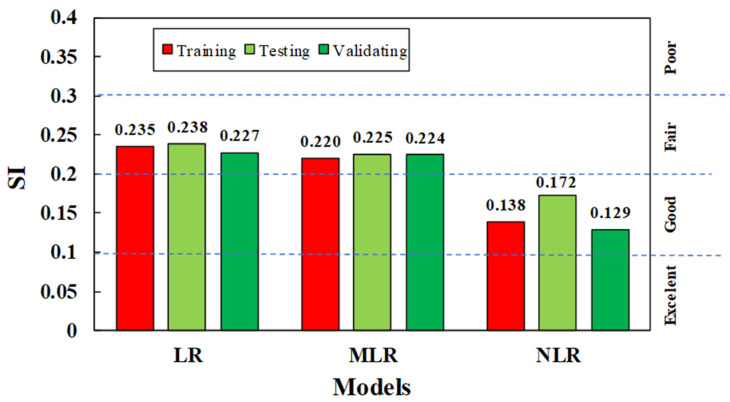
The SI values for all proposed models.

**Figure 23 materials-15-01868-f023:**
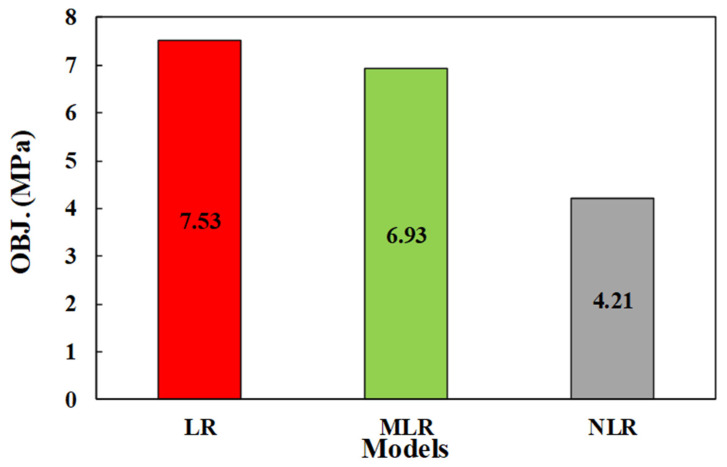
The OBJ values for all proposed models.

**Figure 24 materials-15-01868-f024:**
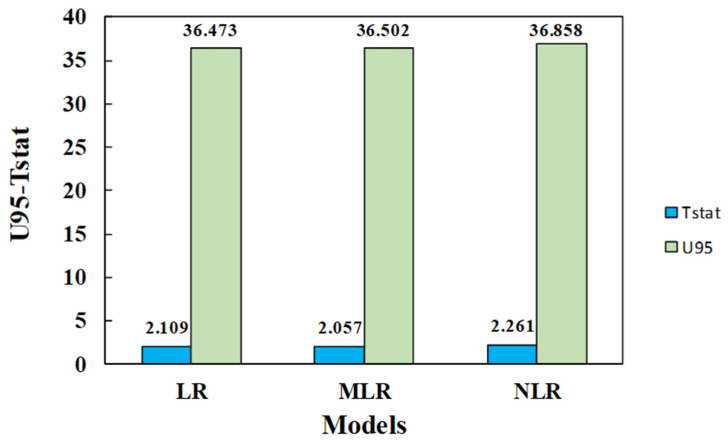
The uncertainty and t-statistic values for all developed models using the testing datasets.

**Table 1 materials-15-01868-t001:** Summary of fly-ash-based geopolymer mortar mixture parameters collected from the literature.

References		[23]	[27]	[32]	[33]	[34]	[35]
**Independent Variables**	FA (kg/m^3^)	909	500–520	625	714	460	600
	SiO_2_ (%)	59.9	43.4–77.2	48	61.92	54.72	65.9
	Al_2_O_3_ (%)	24.7	15.2–33.4	23.1	28.1	27.28	24
	Sand (kg/m^3^)	1173	1375–1430	937.5	1286	1840	750
	NaOH (kg/m^3^)	90	59–145	125	102–178.5	53	110
	Na_2_SiO_3_ (kg/m^3^)	228	124–260	187.5	178.5–255	131	220
	SiO_2_/Na_2_O	2.5	2	3.21	2.88	2.5	2
	H_2_O/Na_2_O	4.83	3.8	6.15	5.21	4.83	3.8
	l/b	0.4–0.6	0.4–0.7	0.5	0.5	0.8	0.6
	NaOH (M), mol/L	44,789	44,911	14	44,852	12-Jan	10
	Curing temperature (°C)	65–80	70	25,30,60	70	65	70
	Curing time (h)	24	24	18	24	24	24
	Age (Days)	7, 14, 28	7	1, 7, 28	1, 7, 14	3, 7, 14, 28	28
**Dependent Variable**	σ_c_ (MPa)	2–15.95	7.2–80	23.7–52	23.1–48.8	7.5–9	9.72

**Table 2 materials-15-01868-t002:** Summary of statistical analysis.

Model Parameters	Statistical Parameters					
	SD	Variance	Skewness	Kurtosis	Max.	Min.
**Fly ash (kg/m^3^)**	136.64	18,670.26	1.64	1.16	909	460
**SiO_2_ (%)**	10.25	105.07	−0.02	−0.69	77	43
**Al_2_O_3_ (%)**	5.06	25.62	−0.61	0.36	33	15
**Sand (kg/m^3^)**	141.88	20,130.33	−0.98	4.13	1840	750
**NaOH (kg/m^3^)**	25.84	667.89	0.47	−0.11	179	53
**Na_2_SiO_3_ (kg/m^3^)**	40.87	1670.7	−0.02	−0.84	293	124
**SiO_2_/Na_2_O**	0.3	0.09	1.4	0.46	3	2
**H_2_O/Na_2_O**	0.64	0.4	1.6	1.6	6	4
**l/b**	0.1	0.01	0.36	−0.76	0.8	0.4
**NaOH(M)**	2.11	4.46	−0.39	−0.23	18	8
**Curing temperature (°C)**	6.85	46.96	−5.24	29.47	80	25
**Curing time (h)**	1.13	1.27	−4.98	22.97	24	18
**Age (Days)**	5.55	30.81	2.69	6.76	28	1
**Compressive strength (MPa)**	18.36	337.01	0.46	−0.61	80	2

**Table 3 materials-15-01868-t003:** Summary of sensitivity analysis.

Sr. No	Input Combination	Removed Parameter	R^2^	MAE (MPa)	RMSE (MPa)	Ranking
**1**	FA, FSO, FAO, S, SS, SH, SO/N, H/N, l/b, M, te, t, Ag	-	0.833	5.2027	6.8256	-
**2**	FSO, FAO, S, SS, SH, SO/N, H/N, l/b, M, te, t, Ag	FA	0.718	7.1929	8.8982	3
**3**	FA, FAO, S, SS, SH, SO/N, H/N, l/b, M, te, t, Ag	**FSO**	**0.5289**	**8.179**	**11.5008**	**1**
**4**	FA, FSO, S, SS, SH, SO/N, H/N, l/b, M, te, t, Ag	FAO	0.8276	5.2765	6.9577	10
**5**	FA, FSO, FAO, SS, SH, SO/N, H/N, l/b, M, te, t, Ag	S	0.8272	5.4084	6.9649	9
**6**	FA, FSO, FAO, S, SH, SO/N, H/N, l/b, M, te, t, Ag	SS	0.7504	6.5679	8.3715	5
**7**	FA, FSO, FAO, S, SS, SO/N, H/N, l/b, M, te, t, Ag	SH	0.7395	7.0099	8.5532	4
**8**	FA, FSO, FAO, S, SS, SH, H/N, l/b, M, te, t, Ag	SO/N	0.8324	5.2663	6.8609	13
**9**	FA, FSO, FAO, S, SS, SH, SO/N, H/N, M, te, t, Ag	H/N	0.8237	5.4286	7.0365	8
**10**	FA, FSO, FAO, S, SS, SH, SO/N, H/N, l/b, M, te, t, Ag	l/b	0.676	7.8737	9.5374	2
**11**	FA, FSO, FAO, S, SS, SH, SO/N, H/N, l/b, te, t, Ag	M	0.8323	5.2461	6.8625	12
**12**	FA, FSO, FAO, S, SS, SH, SO/N, H/N, l/b, M, t, Ag	te	0.8308	5.2474	6.8926	11
**13**	FA, FSO, FAO, S, SS, SH, SO/N, H/N, l/b, M, te, Ag	t	0.8176	5.5436	7.1563	7
**14**	FA, FSO, FAO, S, SS, SH, SO/N, H/N, l/b, M, te, t	Ag	0.8119	5.7051	7.2671	6

## Data Availability

The data presented in this study are available on request from the corresponding author.

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
