# Peer review of "Statistical Methods for Modeling the Compressive Strength of Geopolymer Mortar"

_materials, 2022, doi:10.3390/ma15051868_

Round 1

Reviewer 1 Report

Dear author,
please consider the following remarks.

  1. The article's title should present a genus-differentia definition of the research object. The word "Various" in the title is superfluous. If the word "model" is in the plural in the title, then it is obvious that the models are different from each other.
  2. The structure of the article does not comply with the IMRAD format. Please rename the sections of the article according to the IMRAD format. Use non-IMRAD titles only for subsections, paragraphs, subparagraphs (with their multi-level numbering).
  3. Line 56. It is not clear what the word "this" means. It is recommended to add a noun connecting the previous and the current sentence after the word "this".
  4. Line 58, 65. In a short test fragment, the phrase "On the other hand" is repeated twice. However, the phrases "First hand" are missing.
  5. On line 335, the author claims the compressive strength ranged from 1.53 to 80.4 MPa. At the same time, the author refers to the literature's collected data, which is summarized in Table 1. In contrast to this, in point "a" of the Conclusion, the author claims that strength is as high as 80 MPa. There is no reference to the literature data here (Line 505). This formulation gives the erroneous impression that the author himself established the strength value equal to 80 MPa.
  6. Use the italic font for variables in the body of the article.
  7. The equation is part of the text of the article. The equation must be followed by a punctuation mark, such as a comma or period.
  8. References are not formatted correctly. Use free Mendeley reference manager (or similar software) is recommended. Use MDPI citation style, available for installing to Mendeley software with entering the link http://www.zotero.org/styles/multidisciplinary-digital-publishing-institute. Be sure that you choose the right citation style in MS Word.

Reviewer 3 Report

The author studied “Various models to predict the compressive strength of fly ash-based geopolymer mortar with multiple mix proportions” is very meaningful.. However, 247 experimental data from previous publications were collected and statistically analyzed. It will be difficult to control single factor variables, resulting in large error of the model and difficult to predict experimental results. It is hoped that some experiments can be supplemented for verification. In addition, there are many clerical errors and inconsistent descriptions in the paper, so I hope the author revise it and re-submit it.

Round 2

Reviewer 2 Report

Please find the comments in the file attached.

Reviewer 3 Report

The paper has been revised and can be accepted.

Round 3

Reviewer 2 Report

The paper has been revised and can be accepted for publication.